# Empirical Linkages between Branching, Lending, and Competition: A Study of Pakistani Banks

**Jaleel Ahmed [1], Umar Farooq [2,*], Ahmad A. Al-Naimi [3], Mosab I. Tabash [4] and Krzysztof Drachal [5]**

1 Department of Accounting and Finance, Capital University of Science and Technology (CUST), Islamabad 44000, Pakistan

2 School of Economics and Finance, Xi'an Jiaotong University, Xi'an 710049, China

3 Department of Finance and Banking Sciences, Faculty of Business, Applied Science Private University, Amman 11931, Jordan

4 College of Business, Al Ain University, Al Ain P.O. Box 64141, United Arab Emirates

5 Faculty of Economic Sciences, University of Warsaw, ul. Długa 44/50, 00-241 Warszawa, Poland

* Correspondence: umerrana246@gmail.com

**Abstract:** This study examines the relationship between branching, lending, and competition in Pakistani banks. Due to denationalization, Pakistani banks started to increase their branch networks and change loan and deposit policies. To check the effect of geographic diversification and distance on the performance of banks, the market power of loans and deposits, and the effect of large and medium banks on the performance of small banks, a sample of commercial banks is selected. The study finds that geographic diversification and distance between bank branches and headquarters do not affect the performance of the banks, but geographic diversification of banks in different areas affects the market power of loans and deposits. The results show that medium and large banks do not affect the performance of the small banks because small banks are better performing in the local market. Medium and large banks are affected by the market power of the loans and deposits of small banks. The study recommends an important policy regarding branch management and its effect on bank performance.

**Keywords:** geographic diversification; distance; performance; monopoly market power of loan and deposit; medium and large banks

**JEL Classification:** G21

## 1. Introduction

After privatization in Pakistan, banks increased their branch networks throughout the whole country and changed loan and deposit policies in the local market. However, increasing the number of branches in different geographic areas is not only beneficial but also has a cost. While increasing the branch networks, banks have the advantage of market power in different areas, can collect the information about clients for advancing loans, and can build borrowing relationships with different clients (Degryse and Ongena 2005). Berger and DeYoung (2006) also argue that the banks establishing lending and borrowing relationships with firms face increases in both benefit and cost. When banks open their new branches in different geographic areas, firstly, they collect information related to the financial condition and sources of income of people in that area. When banks open multiple branches in the local market, then banks create a good relationship with firms and attain benefits. Consequently, these banks enjoy the monopolistic deposit and loan power of the market.

In Pakistan, there are many local, cooperative banks which only serve local markets. Small banks are effectively fulfilling the credit need of small businesses in Pakistan. Small banks collect soft information related to the credit needs of the small business (Berger

and Udell 2002). Studies conducted by Berger et al. (2007), Hannan and Prager (2009), Coccorese (2009), and Scott and Dunkelberg (2010) found that when large and medium banks enter the local market to open new branches, then they affect the performance of the small banks in the market.

This study examines the relationships between lending and branching competition, geographic diversification, and distance between bank headquarters and its branches on bank performance, deposits, loan amounts, and monopoly power of deposits. The study also explores the theoretical discussion on market coverage of medium and large banks, performance, monopoly market power of deposits, loans in the market, and the lending strategies of small banks. In Pakistan, the large and medium banks perform better in the loan and deposit market compared to small banks due to low interest rates, and this rate is covered due to the large portion of loans. Small banks only fulfill the loan demand of small businesses.

As a geographic diversification (GD), Pakistan is divided into three regions; the first is the southern region, the second is the central region, and the last is the northern region. The southern region includes Sindh (province) and Baluchistan (province). The central region includes only Punjab (province), and the North region consists of the capital Islamabad, Khyber Pakhtunkhwa (province), Gilgit–Baltistan (province), and Azad Jammu and Kashmir (state). Pakistani banks are geographically diversified in these regions, but large numbers of branches of banks are operating in Punjab. The research objective is to investigate the impact of diversification strategies and distance between bank headquarters and branches on performance, deposits, and loan markets for all banks.

Currently, 31 commercial banks are working in Pakistan and are the members of the PBA (Pakistan Banking Association). According to market capitalization, HBL (Habib Bank Limited), MBL (Meezan Bank Limited), NBP (National Bank of Pakistan), UBL (United Bank Limited), and ABL (Allied Bank Limited) are the largest banks in Pakistan. In terms of size, HBL has over 1700 branches all around the country, MBL has an estimated 890 branches, NBP has 1511 branches, and ABL has a large network of 1425 branches across the country. The total market capitalization of all private banks constitutes almost 45.33% of the total GDP for the year 2021 in Pakistan. According to WDI (World Development Indicators) and the World Bank, the total domestic credit provided by the banks to the private sector was only 15.033% of GDP in the year 2020.

This study tested the agency theory. In the performance of a bank, agency theory always plays a vital role. The shareholders do not connect directly with the organizations, but the managers act in their interest. In managerial firms, the owners and managers are separate, and policies and laws made by the manager may create conflicts. Fama (1980) reported that this problem occurred when most decisions were made solely by executives. The top management or executives think that they make the profit of the firm and perform controls, but they do not own the firm. Due to decision-making power and voting rights, problems are created between shareholders and management, and this agency problem minimizes the efficiency of the firms. Due to more branching strategies adopted by private banks, there are more chances of agency conflicts between banks and shareholders due to low control of headquarters on the branches located in far regions. The branches located in distant areas may not act properly, and therefore, the performance of such branches can be below the mark (Berger and Deyoung 2001). Baysinger and Butler (1985) argued that the board of directors easily removes conflicts from the organization. Banking supervision and risk management of a bank can help in removing the agency conflicts.

We investigate the specific impact of branching, lending, and competition on the performance of banks as it is interesting to note whether more branching strategies improve the performance of banks or not. The private banks invest too much in expanding their business network by establishing more branches even in distant areas. They also follow competitive lending strategies to enhance sales volume. Therefore, it is interesting to explore the impact of such banking strategies on their performance. This study contributes by exploring the role of geographical diversification and more branching strategies in the

performance of banks. Most studies explore the routine determinants of banks, and the literature is scant on such a theme. The study is split into five sections. Section 1 is the introduction, and Section 2 is a review of previous studies. Section 3 discusses the research methodology, and Section 4 explains the results of the study. Section 5 concludes the whole discussion of the study.

## 2. Literature Review and Hypotheses Development

Chiappori et al. (1995) examined the spatial competition in the banking industry, cross-subsidies, localization, and regulation of deposit rates. They studied the outcome of the rules of rates of pay on deposits. Their main objectives were to study the low credit rate due to high competition in the market and different packages of deposits and credit services. The distance between bank branches and their headquarters has been used in earlier studies, but the discussion on the dispersion of branch networks in the region has been ignored. The banks have suffered high observing costs to maintain the branch operations. When the current branch network is large, then the installation of extra branches in the same market negatively impacts the market share.

Li and Greenwood (2004) explained two ideas for the diversification of the firm. The first is diversification of the firms in different areas of possible advantage attaining the economies of scope, but this statement overlooks the opportunity that diversification also increases the performance of the firm because the diversified firm was capable to compete with other firms so that banks are geographically diversified and are capable to compete with the multimarket and experienced banks. However, the study of Brighi and Venturelli (2016) asserts the negative impact of geographic diversification on bank profitability. They analyzed 491 Italian banks and found a negative relationship between geographic diversification and bank performance.

A study on five famous measures of banking market competition and regular and contradictory forecasts of competitive behavior within the country and across the counties was conducted by Carbó et al. (2009). The five measures are the Lerner index, H-statistic, interest margin, HHI (Herfindahl–Hirschman Index), and return on assets, and these are related to each other. Four are positively associated with each other and one is negatively associated. The profitability and multimarket contact in the banking industry of Italy over the period 2002 to 2005 was examined by Coccorese and Pellecchia (2009). They show that diversification strategy is positive and significantly linked with profitability. The multimarket relationship diminishes the competition in the Italian banking sector (Coccorese and Pellecchia 2013). Based on the arguments provided above, they hypothesized that:

**H₁.** *Geographic diversification of banks negatively affects bank performance.*

**H₂.** *The distance between bank branches and their headquarters negatively affects bank performance.*

In the banking industry, information is not only part of the measure of the loan market competition, but other factors like informational, technological, and institutional factors also contribute to decreasing the number of competing creditors and also play the key role in assessing the financial positions of customers. Moreover, the measure of the contestability differs across the border, and parts of the loan market and the supervisors refine their purpose for encouraging competition (Golesorkhi et al. 2019). Geographic proximity to customers represents a main competitive benefit, specifically in the case of a loan for small businesses in terms of transaction costs, information costs, and transportation costs (Dell'Ariccia 2001). Cerqueiro et al. (2009) studied distance, lending decisions, and bank organizational structure. They argued that the banks want to gather information related to the local financial condition and assess the customer on their loan profile. It is an important factor for the customer to overcome unequal information problems related to the bank's proximity. The current technological revolutions such as internet banking and mobile banking have assisted customers, and the bank collects information relating to market entry and minimizing the proximity of banks.

Degl'Innocenti et al. (2017) argued that geographic diversification policies and the increasing number of branches in different geographic areas can increase the current asymmetric difficulties between borrowers and banks. Increasing the number of bank branches causes a reduction in the organization's efficiencies and creates agency costs, but at the same time, the number of bank branches shows the main objective for banks to collect information related to offering loans and deposits to customers. So, increasing the bank branches geographically over the country promotes deposit taking and loan making for banks. When distance increases between the bank's headquarters and its branches, it can damage the relationship of the bank with local firms or industry and impact the loan amounts provided to the client. Small and less diversified banks geographically generally establish relationships with local customers, especially in a high number of small businesses in the market.

**H₃.** *Geographic diversification of banks has a positive impact on the market power of loans, the deposit market, and loan and deposit quantity.*

**H₄.** *Distance between branches of the bank and its headquarters has a negative impact on the market power of loans, the deposit market, and loan and deposit quantity.*

Large banks are assumed to set the interest rates for retail customers. Such rates are uniformly adopted by small banks across the market (Park and Pennacchi 2009). They explain the medium, large, and multimarket banks' effect on small and single-market bank deposits and loan activities. These banks set the rate of deposits and retail loans in the market, but they set rates based on market distribution and market depending on small banks and large and multimarket banks alongside market awareness. Specific wholesale funds on low-cost, medium, and multimarket banks can encourage loan market competition. The cooperation between medium and multimarket banks leads to a declining trend in the lending rate of banks on deposits. Banks offer low deposit rates to their customers. Therefore, single-market banks have an advantage in terms of profits, in the case of the existence of medium and large banks. However, they increase the competition in the loan market between small, single-market banks and multimarket banks.

Petersen and Rajan (2002) argued that the operating medium and large banks are different from single-market banks because the activities of medium and large banks are standardized and decisions to offer loans to the customer based on the financial information of the borrower generally available in the market. Haynes et al. (1999) and Berger et al. (2005) argued that medium and large banks are requiring extra financial histories of the loan candidates of small businesses. However, small and single-market banks can better perform in the local market because they gather better soft information related to the candidates in the local market for advancing the loans (Nuseir and Qasim 2021).

The Italian banking industry decreases the market power of small and single-market banks and increases the competition in the local market of large and multimarket banks (Coccorese 2009). The general impact of geographic diversification on the deposit amount offered by the medium and large banks still has contentions. However, Park and Pennacchi (2009) conjectured that the growth of small and single banks should be measured in the quantity of deposits and deposit rates offered to the depositors as compared to the medium and large banks, but at the same time, the medium and large banks offer extra, multiple services to the customer in the local market and develop a better reputation. So, we can expect the number of deposits of the medium and large banks to increase and decrease the number of deposits of the small and single-market banks in the local market.

Hannan and Prager (2009) studied the relationship between single markets and small banks' profitability in a multimarket banking sample of single-market banks over the period 1993–2003. They argue that the large and multimarket banks offering the prices in the market do not indicate the variation in the local banking system. The small and single-market banks performed better than the large medium and multimarket banks because the single-market banks easily competed in the market. They also explain that small banks have better performance in urban areas rather than the rural areas. Hannan

and Prager (2004) also argued that large and multimarket banks offer a low-interest rate on deposits against small single-market banks.

**H₅.** *The market power for loans and deposits and the performance of banks decrease as the medium and large bank branches increase in the local market.*

**H₆.** *The small banks' quantity of loans and deposits decreases as the medium and large bank branches increase in the local market.*

Extending the discussion of the comparative literature review, the study of Budhathoki et al. (2020) arranged an empirical analysis on the Nepalese banking sector and examine the trend of competition across the banks. They conjectured that Nepalese banks are working under the situation of perfect competition and have shifted from the monopolistic nature of competition to perfect competition. Coccorese and Santucci (2020) assessed the degree of competition across Italian banks and its possible effect on bank size. Their analysis reveals that smaller banks enjoy more competitive advantages in the shape of high market power. Atkins et al. (2022) examined the role of race in bank lending policy in the United States and vowed those black-owned enterprises receive fewer loans as compared to white-owned enterprises. However, this effect was significantly moderated by bank competition as high bank competition affects the lending strategy and removed such distinctions from the market. By using the novel dataset of the Ukrainian banking sector, the study of Pham et al. (2022) asserted that more branching through the establishment of more contact points can enhance the supply of credit. They further reveal that bank diversification strategies help in mitigating the default risks. Wang et al. (2022) investigated the impact of bank deregulation strategy on credit risk in Chinese banks and found that such a strategy augments credit risk. The empirical findings of these studies demonstrate the trend of branching, lending, and competition in other economies of the world.

## 3. Research Methodology

### 3.1. Data Description

The population of this study is based on the banking sector of Pakistan. These banks are under the regulation of the State Bank of Pakistan. The duration period of the study is from 2006 to 2016. Overall, 26 privatized, public, and private banks operate in Pakistan. The selection of banks as a sample is performed based on data available on the bank branches, so 21 banks are selected as a sample size. All the data have been collected through the banks' annual financial reports, PBA (Pakistan Banking Association), and financial statement analysis published by the State Bank of Pakistan.

### 3.2. Methodology

In this study, we used two models to check the geographic diversification, performance, and local market structure. The first model explains the performance and geographic diversification of the banks, and the second model explains the local market structure of the banks.

#### 3.2.1. Geographic Diversification and Performance

The relationship among variables can be presented in the shape of following econometric equation.

$$Y_{it} = \beta + \beta_1 GD_{it} + \beta_2 DIS_{it} + \beta_3 EQTA_{it} + \beta_4 MC_{it} + \beta_5 NPLs_{it} + \beta_6 HHI_{it} + \lambda_i + \varepsilon_{it} \quad (1)$$

where $Y_{it}$ are dependent variables, this study used five dependent variables in separate tenders of the model first is the return on assets ROA, and return on assets is the proxy of performance. ROA is the ratio of net income to total assets. The independent variables are geographic dispersion (GD), and $GD_{it}$ is a measurement of geographic dispersion. This measure reflects the number of the region and the number of branches where the bank operates. Second and third are the Lerner index for deposits and loans, which is the

measure of the monopoly market power of loans and deposits. Fourth is the change in loans from period $t-1$ and $t$, CIL, and last is a change in deposits from period $t-1$ and $t$, CID. In this study, we calculate the GD through the Berry Index as follows:

$$D_i = a - \sum_j \left( \frac{b_{ij}}{\sum jb_j} \right)$$

where $b_{ij}$ is the number of branches related to the bank i in the j region, $j$ = 1 to 3 regions in Pakistan. The Berry index is a measure between 0 and 1, and if the answer is equal to 1 it means the bank is fully diversified. Distance (DIS) is calculated between bank branches and their headquarters. It is calculated as the distance between two exact locations by use of zip codes. The weighted distance is measured as the total distance in kilometers between bank branches and their headquarters, the weight being the bank branches in a region as a share of the total branches of the same bank. Degl'Innocenti et al. (2017) asserted that geographic diversification and distance affect the bank's performance and monopoly market power of loans and deposits because when banks open their branches in different geographic and distant areas, it affects the performance of the banks and monopoly market power of loans and deposits. In particular, the increased area affects the market performance and lending capacity of the banks and stems from weak control of distant branches and more chances for agency conflicts.

Market coverage (MC) is calculated as the total branches of banks i operating in all regions at time t divided by the total branches of all banks. Non-performing loans (NPLs), it is the indicator of risk management quality. NPLs are the ratio of non-performing loans to total assets. The equity-to-total-assets (EQTA) ratio shows the capitalization of the bank. EQTAit is the ratio of equity to total assets. Last, the Herfindahl–Hirschman Index (HHI) shows the value of the Herfindahl–Hirschman Index. HHI is used to measure market concentration. HHI is calculated as the sum of the square of the market share of each bank operating.

$$HHI = B_1^2 + B_2^2 + B_3^2 + B_4^2 + \cdots B_n^2$$

B shows the bank market share, while n is total number of banks in a market.

### 3.2.2. Local Market Structure

The second model examines the variation in the local market structure and increases medium and large banks' market coverage in all regions affecting the performance of loans amount, deposits, and Lerner index of deposits and loans of small banks (small banks means these banks are operating in some cities), according to Coccorese (2009) and Hannan and Prager (2009). The following model will be the second model:

$$Y_{it} = \beta_\circ + \beta_1 MCL_{it} + \beta_2 HHI_{it} + \beta_3 HHI_{it} * MCLarge_{it} + \beta_4 MC_{it} + \beta_5 NPLs_{it} + \beta_6 EQTA_{it} + \lambda_i + \varepsilon_{it} \qquad (2)$$

where $Y_{it}$ are dependent variables; this study uses five dependent variables in separate applications of the model. The first is the return on assets ROA; the second is (DPTA) the deposit over total assets; the third is (LNTA) loans over total assets; and the fourth and fifth are the Lerner indices for deposits and loans. The independent variables are MCL, which is fractioned between number of branches owned by medium and large banks divided by overall branches in each region at the time $t$; market coverage (MC); non-performing loans (NPLs); equity-to-total-assets (EQTA) ratio; and last, the Herfindahl–Hirschman Index (HHI). The economic forces increase by the medium and large bank branches in the local market. The λ fixed effect model and $\varepsilon_{it}$ show the error term in the model.

### 3.2.3. Monopoly Market Power

Banking literature mostly uses the Lerner index to measure the monopoly market power of banks. The following authors used the Lerner index for loans and deposits to

measure the monopoly market power: (Degl'Innocenti et al. 2017; Forssbæck and Shehza 2014; Koetter et al. 2012; Carbó et al. 2009). The formula of LERNER is as follows:

$$LERNER_{it} = (P_{it} - MC_{it})/P_{it}$$

where $MC_{it}$ is the marginal cost of the firm, and $P_{it}$ shows the market price of the firm. This study investigates the effect of banking structure on the market power in deposit and loan markets and calculates the Lerner index of deposits and Lerner index of loans.

Lerner Index for Loan LL

The Lerner index for a loan measures the market power of the loan. *LL* is a Lerner Index for loans, and it is calculated as

$$LL_{it} = \left[\frac{PL_{it} - MCL_{it}}{PL_{it}}\right]$$

where $PL_{it}$ is interest income, and $MCL_{it}$ is the marginal cost of loans.

Lerner Index for Deposits LD

The Lerner index for deposits is measure the market power of deposits. *LD* is a Lerner Index for deposit, calculated as

$$LD_{it} = \left[\frac{PD_{it} - MCD_{it}}{PD_{it}}\right]$$

where *PD* is interest expenses on deposits and approximates the average costs, and *MCD* is the marginal cost of deposits. This shows the percentage markup that a bank is able to cover over its marginal cost. It also exhibits the capacity of banks to set the interest income over its interest expenses. This study follows Degl'Innocenti et al. (2017), Forssbæck and Shehza (2014), and Williams (2012) to drive the translog cost function. The following formulas are *MCL* and *MCD*:

$$MCL_{it} = \left[\gamma^L + \gamma^{LL} LnLoans_{it} + \gamma^{LD} lnDep_{it} + \sum \gamma^{hl} lnWH_{it} + tLT_{it}\right]\left(\frac{TC_{it}}{Loans_{it}}\right)$$

$$MCD_{it} = \left[\gamma^D + \gamma^{LD} lnDep_{it} + \sum \gamma^{hl} lnWH_{it} + tLT_{it}\right]\left(\frac{TC_{it}}{deposits_{it}}\right)$$

TC is the sum of other administrative expenses and personnel expenses, $lnDep_{it}$ is the log of deposits, and *lnLoans* is the log of loans. *lnWh* is the labor and capital prices, labor is the ratio of other administrative expenses to total assets, and the capital price is the ratio of the number of employees to personnel expenses. Here, capital means "human capital" costs which is a costs per employee determined as sum of personal expenses including salaries and other benefits divided by the number of employees. This is a rough calculation for measuring how much money each employee generates for the bank. LT is a log of total assets owned by banks, and TC is a transaction cost while subscripts *i* is cross-section and *t* is time effect.

## 4. Results and Discussion

### 4.1. Descriptive Statistics

Table 1 shows the descriptive analysis. Descriptive statistics help to define the basic characteristics of the data and present a simple and easy summary of the sample and measure. Descriptive statistics include mean median, minimum, and maximum standard deviation.

**Table 1.** Descriptive Statistics.

| Variables | Mean | Median | Maximum | Minimum | Range |
|---|---|---|---|---|---|
| ROA | 0.0030 | 0.0100 | 0.0400 | −0.0900 | 0.1300 |
| LL | 0.680 | 0.692 | 0.770 | −0.183 | 0.188 |
| LD | 0.115 | 0.105 | 0.376 | −0.360 | 0.412 |
| CIL | 0.2046 | 0.1200 | 0.4210 | −0.0100 | 0.5210 |
| CID | 0.2209 | 0.1600 | 2.7500 | −0.2700 | 3.0200 |
| LNTA | 0.5092 | 0.4800 | 0.755 | 0.1700 | 1.3800 |
| DPTA | 0.7394 | 0.7600 | 0.9100 | 0.2900 | 0.6200 |
| GD | 0.6039 | 0.6300 | 0.6700 | 0.3200 | 0.3500 |
| LDIS | 6.6162 | 6.8000 | 700.000 | 5.6500 | 1.3500 |
| MCIT | 0.0549 | 0.0300 | 0.2600 | 0.0000 | 0.2600 |
| HHI | 0.0208 | 0.0000 | 0.2100 | 0.0000 | 0.2100 |
| NPLS | 0.1467 | 0.1000 | 0.7400 | 0.0100 | 0.7400 |
| EQTAIT | 0.1014 | 0.0700 | 0.5000 | −0.0300 | 0.5300 |
| MCL | 0.4707 | 0.2500 | 0.8800 | 0.1200 | 0.7600 |

Note: This table shows the descriptive stats in the form of mean, median, maximum, and range.

The summary statistics about ROA show the mean value of 0.0030 in the overall Pakistani banking industry. This value shows that the average return on assets of banks is 0.3%. The minimum value of return on assets is −0.09, which means some banks' return on assets at that time is −9%, but the maximum value of return on assets is 0.04, which means the maximum return on assets of banks is 4%. LL is the abbreviation of the Lerner index of loans, and it is used to calculate the monopoly market power of loans. The mean value of LL depending on the variable is 0.680, while the maximum value of the Lerner index of loans is 0.770, demonstrating the market power of banks. The minimum value of the Lerner index of loans is −0.183, which means, at that time, some banks' loan amounts were negative. LD is a Lerner index of deposits and shows the monopoly market power of deposits. The minimum value of the Lerner index of deposits is −0.360, which means the deposit amount of some banks at that time was negative. The maximum value of the Lerner index of deposits is 0.376, while the average value LD of banks in Pakistan is 0.115. The average value of change in loan CIL of banks in Pakistan is 0.2046. It means that every year, 20.46% of loan amounts increase. The maximum value of change in the loan is 0.4210; it means some banks' loan amounts increase 42.100% from the previous year. The minimum value of CIL is −0.0100, which means some banks' loan amounts decreased from the previous year to the current year by −1%. The mean value of change in deposits is 0.2209 in the overall Pakistani banking industry. It means that, every year, there is a 22.09% increase in deposit amounts. The maximum value of change in deposits is 2.7500 because some banks cover large markets, so they collect a large portion of deposits. The minimum value of change in deposits is −0.2700. LNTA means the loans over total assets, and the mean value of overall banks in Pakistan for LNTA is 0.5917, which means banks have 59.17% of loans provided to the customer, while the maximum and minimum values are 0.755 and 0.1700. The average value of DPTA in Pakistan's banking industry is 0.7394, which means banks have 73.94% of assets financed by deposits. The maximum and minimum values of deposits over total assets (DPTA) are 0.9100 and 0.2900.

The average value of GD, geographic dispersion, of banks is 0.6039 in Pakistan overall. The minimum value of GD is 0.3200 because banks are diversified in different areas, so some banks are less diversified. The maximum value of GD is 0.6700, which means some banks are highly diversified. LDIS means the distance from headquarters to bank branches, and the average distance value of a bank's headquarters to its branches is 6.6162 km. The

maximum value of the distance is 700.000 because banks are operating in three different regions, so some branches are operating at short distances and some branches operate at a large distance. The minimum distance from the bank headquarters to the branches is 5.6500. $MC_{it}$ is the market coverage of Pakistani banks in all three regions. The average value of $MC_{it}$ is 0.0549, which means the average market covered by banks is 5.49%. The maximum value of $MC_{it}$ is 0.2600, so some banks are covering the 26.00% market. The minimum value of $MC_{it}$ is 0.000, which means some banks are not operating in all three regions, so they do not cover the market. The average value of the HHI Herfindahl–Hirschman Index is 0.0208 in the Pakistan banking industry. The maximum and minimum values of the Herfindahl–Hirschman Index are 0.2100 and 0.000. NPLs are non-performing loans, and the average value of NPLs in Pakistan banks is 0.1469, which means 14.69% bad debts against the loan amount. The maximum value of non-performing loans is 0.7400, which means some banks have 74% bad debts against loans, while the minimum value of NPLs is 0.0100; it means some banks have 1% bad debts against loans. The mean value of equity over total assets $EQTA_{it}$ in the Pakistan banking industry is 0.1014, which means 10.14% of assets are generated against equity. The maximum value of equity over total assets is 0.5000, which means some banks are 50% assets generated against equity, while the minimum value of equity over total assets $EQTA_{it}$ is −0.0300; it means some banks have −3% assets generated against equity. The average value of $MClarge_{it}$ is 0.4707, which means the large and medium banks cover 47.07% of the market in Pakistan. The minimum and maximum values of large and medium banks $MClarge_{it}$ are 0.1200 and 0.8800.

*4.2. Correlation Analysis*

Correlation shows the association among the variables. Correlation also shows whether the relationship between two variables is strongly or weakly correlated. In Table 2, the return on assets is positively correlated with the Lerner index of loans and the Lerner index of deposits, 0.0115 and 0.1422, respectively. Return on assets is negatively correlated with change in loans, and loans over total assets and their coefficient values are −0.0289 and −0.5573, respectively. The change in loans may be negative (decrease in loan volume), and underlying banks may suffer from low return on assets due to less sale volume (advancing the loans to the customer is the sale of banks). Changes in deposits and deposits over total assets are positively correlated with return on assets of 0.1422 and 0.2865. Geographic dispersion and distance are negatively correlated with return on assets and their correlation values are −0.0622 and −0.0355, respectively. This negative correlation can be explained as when the distance increases between bank branch and headquarters, it reduces the return on assets due to agency issues. Return on assets is positively correlated with market coverage and HHI at 0.4546 and 0.3682; when the assets of banks are increasing, then investment opportunities also increase so the market coverage and market share of the banks increase. Equity-to-total-assets ratio and NPLs are negatively correlated with return on assets at −0.2061 and −0. 6046. respectively. Return on assets is positively correlated with a market coverage of large and medium banks at 0.3749 because when large and medium banks cover the large market then the return on the asset is high, and these are positively correlated. The Lerner index of loans is highly positively correlated with the Lerner index of deposits at 0.9111 because when the customer deposits are high, then the bank has the opportunity to invest these amounts in different projects and provide the loans to customers and receive a high amount against loans, and so these are highly correlated with each other.

The Lerner index of loans is positively correlated with geographic dispersion, distance, market coverage, NPLs, HHI, and market coverage of large and medium banks at 0.1903, 0.1880, 0.1970, 0.1024, 0.1785, and 0.2059, respectively, but the Lerner index of loans is negatively correlated with equity over total assets at −0.1357. The Lerner index of the deposit is negatively correlated with change in loan, −0.2474, and change in deposits, −0.2432. Distance, 0.0819, and geographic dispersion, 0.1488, are positively correlated with the Lerner index of deposits. HHI, 0.4325, and market coverage, 0.4360, are positively

correlated with the Lerner index of deposits; when the market share and market coverage are high, then the deposits of the banks are high, so these are highly correlated with each other. Equity over the total asset is negatively correlated with the Lerner index of deposit, −0.0750, and market coverage of medium and large banks is positively correlated at 0.3180 with the Lerner index of deposit because medium and large banks cover a large market with respect to small banks.

**Table 2.** Correlation Analysis.

|  | ROA | LL | LD | CIL | CID | LNTA | DPTA | GD | LDIS | MCIT | HHI | NPLS | EQTAIT | MCL |
|---|---|---|---|---|---|---|---|---|---|---|---|---|---|---|
| ROA | 1.000 | | | | | | | | | | | | | |
| LL | 0.011 | 1.000 | | | | | | | | | | | | |
| LD | 0.142 | 0.911 | 1.000 | | | | | | | | | | | |
| CIL | −0.028 | −0.206 | −0.247 | 1.000 | | | | | | | | | | |
| CID | 0.003 | −0.192 | −0.243 | 0.762 | 1.000 | | | | | | | | | |
| LNTA | −0.557 | 0.016 | −0.003 | −0.140 | −0.191 | 1.000 | | | | | | | | |
| DPTA | 0.286 | 0.008 | −0.051 | −0.156 | −0.075 | −0.412 | 1.000 | | | | | | | |
| GD | −0.062 | 0.190 | 0.148 | −0.026 | −0.030 | 0.011 | −0.079 | 1.000 | | | | | | |
| LDIS | −0.035 | 0.188 | 0.081 | 0.113 | 0.082 | −0.184 | 0.010 | 0.635 | 1.000 | | | | | |
| MCIT | 0.454 | 0.196 | 0.436 | −0.153 | −0.151 | −0.172 | 0.272 | 0.019 | −0.128 | 1.000 | | | | |
| HHI | 0.368 | 0.178 | 0.432 | −0.114 | −0.114 | −0.089 | 0.153 | 0.069 | −0.070 | 0.963 | 1.000 | | | |
| NPLS | −0.604 | 0.102 | 0.062 | −0.182 | −0.119 | 0.761 | −0.483 | −0.099 | −0.308 | −0.266 | −0.181 | 1.000 | | |
| EQTAIT | −0.206 | −0.135 | −0.075 | 0.238 | 0.210 | 0.266 | −0.626 | 0.217 | 0.128 | −0.234 | −0.111 | 0.217 | 1.000 | |
| MCL | 0.374 | 0.205 | 0.317 | −0.206 | −0.197 | −0.084 | 0.429 | −0.116 | −0.256 | 0.652 | 0.529 | −0.196 | −0.424 | 1.000 |

Note: The dependent variables are return on assets—ROA; Lerner index for deposits—LD; Lerner index for loans—LL; change in loan—CIL; change in deposits—CID; loan over total assets—LNTA: deposit over total assets—DPTA. The independent variable GD is the geographic diversification of the banks; LDIS is the distance between the bank headquarters and its branches; $MC_{it}$ is the market coverage; HHI is the Herfindahl–Hirschman Index; NPLs are non-performing loans; Equity-over-total-assets—$EQTA_{it}$; MClarge is the market coverage of large and medium banks.

Distance has a high positive correlation with geographic dispersion, 0.6351, because when banks open their branches in different areas, then the distance of the banks' headquarters to its branches increase, so these are highly correlated. GD, geographic dispersion, has a negative correlation with NPLs, −0.999, and market coverage of medium and large banks, and all other variables are positively correlated with GD. Distance has a positive correlation with $EQTA_{it}$ equity over total assets, 0.1285, but all the other variables are negatively correlated with distance.

Market coverage has a highly positive correlation with HHI, the Herfindahl–Hirschman Index, 0.9635, because when banks cover a large market then the market share of the banks is high; when the market share of the banks is high, then the value of HHI, the Herfindahl–Hirschman Index, is high, so these are highly positively correlated. $Mc_{it}$ has a high positive correlation with the market coverage of medium and large banks, 0.6521, because medium and large banks cover the large markets in Pakistan, and small banks cover a small part of the market, while medium and large banks cover the large market and the $MC_{it}$ market coverage is high, and all other variables are negatively correlated with $MC_{it}$.

4.2.1. Results of Regression Model 1

Table 3 shows that geographic diversification and distance affect the performance of banks, and the market power of loans and deposits. The dependent variable is ROA, return on assets, and we used the regression fixed-effect model. The value of R-squared is 0.545754 and shows that the model has strong explanatory power. The variables of geographic diversification, distance, market coverage, and HHI are insignificant.

Geographic diversification and distance do not affect the performance of the banks in Pakistan because Pakistan banking regulations are very strict. The State Bank of Pakistan monitors all the activities of the banks in Pakistan, and the SBP's responsibility is that banks provide all the activities in all the geographic areas and meet the customer needs, so the bank's headquarters controls all the branches in Pakistan. (These are the State Bank of Pakistan regulations). Equity over total assets has a positive significant impact on the performance of the banks; it means if EQTAit increases, then the performance of the banks also increases, and it is a direct relationship between performance and equity over

total assets. This is consistent with the result of the study by Degl'Innocenti et al. (2017). Non-performing loans are a negative and significant impact on a bank's performance. If the NPLs of the banks increase, then the performance of the banks decreases. This is an indirect relation between performance and NPLs.

**Table 3.** Performance and Monopoly Market Power of all Banks.

| Variables | ROA | LL | LD | CIL | CID |
|---|---|---|---|---|---|
| GD | −0.001 | 0.3403 *** | 0.4103 * | 0.475 | 0.864 |
| | [0.053] | [0.4936] | [0.2219] | [1.543] | [1.082] |
| LDIS | 0.0361 | 0.0299 | 0.1342 | −2.365 ** | −1.555 * |
| | [0.043] | [0.0804] | [0.1804] | [1.243] | [0.871] |
| EQTAIT | 0.03 * | −0.0589 ** | −0.1341 ** | 1.34 *** | 0.814 *** |
| | [0.016] | [0.0303] | [0.0681] | [0.418] | [0.293] |
| MCIT | −0.03 | 0.9703 *** | 2.1829 *** | −10.469 *** | −5.927 ** |
| | [0.152] | [0.2949] | [0.6620] | [4.054] | [2.842] |
| NPLS | −0.069 *** | −0.0338 | −0.1076 * | −1.13 *** | −0.478 |
| | [0.015] | [0.0290] | [0.0652] | [0.447] | [0.314] |
| HHI | 0.049 | −0.9644 *** | −1.6772 *** | 10.577 *** | 5.832 ** |
| | [0.143] | [0.2757] | [0.6189] | [4.013] | [2.813] |
| Constant | −0.226 | −0.3581 | −1.0756 | 15.908 ** | 10.173 ** |
| | [0.257] | [0.4936] | [1.1079] | [7.560] | [5.299] |
| Observation | 202 | 201 | 201 | 185 | 185 |
| R-squared | 0.5458 | 0.5588 | 0.5085 | 0.4142 | 0.3768 |
| Adj. R-squared | 0.4782 | 0.4929 | 0.4350 | 0.3178 | 0.2742 |

Note: this table shows the coefficient of Equation (1). The dependent variables are return on assets; Lerner index for deposits—LD; Lerner index for loans—LL; change in loan—CIL; change in deposit—CID. The independent variable GD is the geographic diversification of the banks. LDIS is the distance between bank headquarters and its branches. EQTAit is equity over total assets; MC—market coverage; NPLs—non-performing loans; HHI—Herfindahl–Hirschman Index. *** $p < 0.01$, ** $p < 0.05$, * $p < 0.1$.

The geographic diversification strategies have a positive and significant impact on the monopoly market power of loans (Lerner index of loans). The analysis reveals the acceptance of Hypothesis 3 (H3). In the case of the Lerner index for loans, geographic diversification is affected because some banks provide the loan to a customer at a minimum level of interest rate. Park and Pennacchi (2009) suggested that the banks in the multimarket encourage the competition for loans through fixed low-interest-rate loans. The diversification of bank branches in different geographic areas can enhance the amount of loans. Equity over the total asset and HHI have a negative and significant impact on the monopoly market power of the loan and Lerner index of loans. This negative value implies that if EQTAit and HHI are decreasing, then the Lerner index of loans increases, showing an inverse relation between EQTAit, HHI, and the Lerner index of loans. Market coverage has a positive and significant effect on the Lerner index of loans, suggesting that the market coverage of banks has a direct effect on monopoly market power of loans. The value of R-squared is 0.5588, showing that the model has strong explanatory power. The value of Adj. R-squared is 0.4929 or 49.29%, implying the degree of goodness of fit of the model.

The geographic diversification strategies have a positive and significant impact on the monopoly market power of deposits (Lerner index of deposits). If GD increases, then the Lerner index of deposits also increases because the bank offers a high-interest rate on deposits in the market, so these are directly related to each other. Studies by Degl'Innocenti et al. (2017), Hannan and Prager (2009), and Park and Pennacchi (2009) support our result that large banks enjoy large amounts of funding benefits. Banks can increase their branches in different areas and collect large amounts from different channels, not only from customer deposits in the banks. Equity over total assets $EQTA_{it}$ is a negative and significant effect on the Lerner index of deposits. If the amount of customer deposits increases, then the equity over total assets decreases. Market coverage has a positive and significant impact on the Lerner index of deposits; it means if market coverage of banks increases, then the deposits of banks increase. NPLs and HHI have a negative and significant impact on the Lerner

index of deposits. If the value of non-performing loans decreases, then the amount of customer deposits increases. The value of R-squared is 0.5085 and shows that the model has strong explanatory power. The value of Adj. R-squared is 0.4350, which shows goodness of fit of the model.

The value of R-square is 0.4142 shows that the model has strong explanatory power. The value of Adj. R-square is 0.3178 that the 31.78% variation between the independent variable and dependent variable. Distance has a negative and significant impact on change in loans. It suggests that the distance between bank branches and their headquarters has decreased the growth of loans. If the distance between bank branches and their headquarters increases, then the growth of loans decreases. Equity over total assets has a positive and significant impact on change in loans. Market coverage and non-performing loans are negative and have significant effects on changes in loans. Market coverage decreases then the number of loans is increasing if the quantity of loans increases, then NPLs also decrease; these are indirectly related to the number of loans. The Herfindahl–Hirschman Index (HHI) has a positive and significant impact on the number of loans. The distance has a negative and significant effect on the deposits; it means a large distance between bank branches, and headquarters decreases the number of customer deposits in the banks. Chiappori et al. (1995) explained that large branches increase the highly competitive pressure and decrease the market power of a single branch in the deposits market. On customer deposits, banks offer a high-interest rate. EQTAit and HHI have a positive and significant impact on the deposits. The market coverage has a negative and significant impact on the deposits. The value of R-square is 0.3768 and shows that the model has goodness of fit. The value of Adj. R-square is 0.2742, which is 27.41% variation between the independent variable and dependent variable.

4.2.2. Results of Regression Model 2

Table 4 shows the medium and large banks' performance and the impact of the monopoly market power of loan and deposits. Past studies (Hannan and Prager 2009; Coccorese 2009) showed that the small and single-market banks are performing better than the large, medium, and multimarket banks because the single-market banks easily compete in the market. These studies also concentrated on the performance of rural and urban banks. They explain that small banks have better performance in urban areas rather than rural ones.

Our results show that the small banks are performing better than the large and medium banks because all banks follow Pakistan banking regulations (Ali and Puah 2018). Small bank branches are fewer in the different areas rather than the large and medium banks, so small banks easily control all the activities in the banks, but large and medium banks are better performing in the loan and deposits market because the small banks do not fulfill the loan demand of the large businesses, but small banks are fulfilling the loan demand of the small businesses. In the loan and deposits market, the monopoly of large and medium banks is because the large and medium banks easily meet the credit needs of the large businesses and customers. Large and medium bank branches are available in all the cities of Pakistan and offer all the facilities to the customer. Degl'Innocenti et al.'s (2017) results show that medium and large banks are better performing than small banks because these banks do not share the same clients. Medium and large banks have capacity to deal with wealthy and corporate customers and are less likely to deal with customers having low creditability and therefore have better performance.

**Table 4.** Regression Analysis for Medium and Larger Banks.

| Variables | ROA | LL | LD | LNTA | DPTA |
|---|---|---|---|---|---|
| MCLARGEIT | 0.005 | 0.0873 * | 0.1308 | 0.389 ** | 0.209 ** |
| | [0.026] | [0.0501] | [0.1123] | [0.171] | [0.107] |
| HHI | −0.150 | 4.18024 *** | 4.9590 * | 0.159 | 1.506 |
| | [0.662] | [1.2494] | [2.8033] | [4.299] | [2.694] |
| HHI*MCLARGEIT | 0.177 | −5.6681 *** | −7.4028 ** | 1.323 | −2.532 |
| | [0.682] | [1.2855] | [2.8843] | [4.426] | [2.773] |
| MCIT | 0.006 | 1.08114 *** | 2.4128 *** | −0.020 | 1.025 * |
| | [0.151] | [0.2906] | [0.6520] | [0.980] | [0.614] |
| NPLS | −0.069 *** | −0.0369 | −0.1124 * | 0.037 | 0.102 |
| | [0.016] | [0.0301] | [0.0676] | [0.104] | [0.065] |
| EQTAIT | 0.028 * | −0.0681 ** | −0.1489 ** | 0.069 | −0.375 *** |
| | [0.017] | [0.0314] | [0.0703] | [0.108] | [0.068] |
| Constant | 0.008 | −0.0149 | −0.0310 | 0.288 ** | 0.618 *** |
| | [0.580] | [0.0284] | [0.0636] | [0.096] | [0.060] |
| Observation | 150 | 150 | 150 | 140 | 140 |
| R-squared | 0.543 | 0.551 | 0.501 | 0.796 | 0.802 |
| Adj. R-squared | 0.475 | 0.484 | 0.426 | 0.766 | 0.772 |
| Durbin–Watson | 1.538 | 1.236 | 1.61 | 1.388 | 1.47 |

Note: this table shows the coefficient of Equation (2). The dependent variables are return on assets; Lerner index for deposits—LD; Lerner index for loans—LL; loan over total assets—LNTA; deposit over total assets—DPTA. The independent variable MClarge is the market coverage of large and medium banks; HHI is the Herfindahl–Hirschman Index; NPLs are non-performing loans; $MC_{it}$—market coverage; Equity over total assets—$EQTA_{it}$. *** $p < 0.01$, ** $p < 0.05$, * $p < 0.1$.

Market coverage of medium and large HHI is a positive and significant impact on the monopoly market power of loans and deposits because when medium and large banks are operating in different cities, and these banks offer a large number of loans to the businesses and collect a large number of deposits of the large businesses and other customers at the high-interest rate, but the small banks are not offering a large amount of loans to the large and medium businesses at a low-interest rate, so the small banks are not competing with the large banks. The value of the R-square Lerner index of deposits and loans are 0.501 and 0.551, and the values of Durbin–Watson of Lerner index of deposits and loans are 1.610 and 1.236. The interaction of HHI and MCLARGE has a negative and significant effect on the Lerner index loans. This negative effect reveals that larger banks face low competition regarding loan advancement. $Mc_{it}$ has a positive and significant impact on the monopoly market power of loans (Berger and Udell 2002; Berger et al. 2005; Uchida et al. 2008); small banks are more efficient to meet the loan needs of small businesses because small banks are gathering better soft information related to the credit of the small business rather than the large businesses. Mostly in the USA, small banks are better performing in the lending strategies because mostly small banks operate in the USA. The US banking system is different from the Pakistani banking system because, in Pakistan, the State bank of Pakistan monitors and controls the banking activities. In Pakistan, the large and medium banks are better for performing loans and deposits rather than the small banks because in Pakistan mostly large and medium banks cover the market and offer credit at a lower interest rate to the business. So, in Pakistan, the large and medium banks perform better.

**5. Conclusions**

This study examined the relationship between branching, lending, and competition in Pakistani banks. Pakistan's government made the privatization commission on 22 January 1991 to privatize the banking sector. This privatization policy adopted by the Government of Pakistan creates high uncertainty among banking institutions, and they started to adopt dynamic strategies relating to the new establishment of banks and loan advancement. Given that, the current study aims to explore the empirical relationship between branching, lending, and competition and how such strategies affect the performance of banks. The panel data and fixed effect model were used for analysis. In the current study, we develop

the two models; the first model analyzes the performance and monopoly market power of all banks, and the second model examines how the medium and large banks affect the performance and monopoly market power of the loans and deposits of small banks. The results of the first model show that geographic diversification has an insignificant effect on bank performance. Increased distance between the bank headquarters and its branches does not affect bank performance because the bank headquarters monitors the activities of its branches (Al-Chahadah et al. 2020). The geographic diversification and distance between the bank's headquarters and its branches are significant effects on the Lerner indexes for loans and deposits. The Lerner indexes for deposits and loans show the monopoly market power of the banks so some banks better perform in the loan and deposit market in different areas. Sometimes banks are offering low interest rates on loans in some areas where competition is high between banks, so geographic diversifications significantly affect the performance.

The results of the second model show that the medium and large banks do not affect the performance of small banks because all banks follow Pakistani banking regulations. Small bank branches are fewer in the different areas rather than the large and medium banks, so small banks easily control all the activities in the banks, which is why small bank performance is better, but in the monopoly market power of loans and deposits, the large and medium banks perform better because the small banks do not fulfill the credit needs of the medium and large businesses, so large and medium banks provide the loans to medium and large businesses.

The current analysis advises an important policy regarding the role of banking diversification on banking performance. It provides policies to both small and medium banks on how more branching and diversification impact the performance of banks. The policy officials can enhance the performance of banks by more focus on branch control and extending loans. Despite the policies, the current analysis has some limitations as it does not consider some other important factors, e.g., the age of banks, management structure, and political affiliations. Moreover, the results of the current analysis cannot be generalized for other economies of the world as each country has a different setup of the banking sector. Future studies can be arranged by extending the analysis to other economies and involving the mentioned factors.

**Author Contributions:** J.A.: Conceptualization, Data curation, Writing—Original draft preparation; U.F.: Writing—Original draft, methodology, Supervision; A.A.A.-N.: Data curation, Reviewing and Editing, Methodology; M.I.T.: project administration; review and editing; investigation; K.D.: formal analysis, Conceptualization, Software. All authors have read and agreed to the published version of the manuscript.

**Funding:** This research received no external funding.

**Data Availability Statement:** Data were collected from the published reports of The State Bank of Pakistan.

**Conflicts of Interest:** The authors declare no conflict of interest.

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
