# Peer review of "Empirical Linkages between Branching, Lending, and Competition: A Study of Pakistani Banks"

_economies, doi:10.3390/economies11050145_

Round 1
Reviewer 1 Report (Previous Reviewer 3)
The research topic is interesting – whether branching structure (distance) influences bank lending policies. Then there was somehow unrelated problem of bank competitive situation (monopoly power), geographic dispersion (branching) and then the brief discussion on the situation of small and large bank. The first version of the reviewed paper was chaotic and did not explain in a responsible manner the links suggested in the title (abstract). The second version does not clarify anything, gives only some additional irrelevant information and is equally chaotic.
The Authors write many pages about less important issues, while the rationale for model building is very brief, and the description of variables is in the footnote! Analyzing the regression results, the Autor interpret statistically insignificant variables
After doing the second review, I think that the construction of the model is incorrect. To research bank monopoly power in lending/deposit market, some customer’ data are required and not only the banking data. The model for loan/deposit growth is interesting, but only marginally described in the paper.
I advise the authors – again - to the research only one basic question, and not try to solve all financial problems (concertation, efficiency, market power, loan/deposit growth, agency theory, etc.) in one paper – it does not work. I have suggested this in the previous review (to reduce the number of hypothesis) but the authors are motivated by a mission to give “a universal advice” to the government, and not to research correctly one issue/hypothesis.
In its current chaotic form I do not see any value added in publish of this paper - with some interesting remarks about the Pakistani banking sector, but with incorrect model with random and unconvincing conclusions.
For example - the author asserts: “geographic diversification and distance between bank branches and headquarters do not affect the performance of the bank” but in the panel model we have ROA, which is positively affected by bank capital level and negatively by NPL, which does not tell us anything about competition and branching. Similarly: “ this study tested the agency theory” by stating in 5 sentenced that this is an important problem. It is not testing, it is an observation.
Author Response
Response Letter
Journal Name: Economies
Manuscript Title: Branching, Lending, and Competition: An Empirical Study of Pakistani Bank
Comments to Editor,
We are thankful for your continuous efforts and bridging the relationship between us and valuable reviewers. I want to let you know that we have carefully considered the points raised by reviewers and incorporated them in body of paper. We have highlighted the corrections with color. We are also grateful to anonymous reviewers who have provided the valuable suggestions to modify the paper.
Response to Reviewers
|
Sr no. |
Comment |
Response |
Page no. |
|
Reviewer#1 |
|||
|
1 |
I advise the authors – again - to the research only one basic question, and not try to solve all financial problems (concertation, efficiency, market power, loan/deposit growth, agency theory, etc.) in one paper – it does not work. I have suggested this in the previous review (to reduce the number of hypothesis) but the authors are motivated by a mission to give “a universal advice” to the government, and not to research correctly one issue/hypothesis. |
Thanks for your kind suggestion. Regarding the narrow down of research, I apologize as this title comes under a project and its necessary to test all underlying pack of variables. However, I assure you that we have inserted a great effort to improve the quality of communication and other parts of paper. |
N/A |
|
2 |
For example - the author asserts: “geographic diversification and distance between bank branches and headquarters do not affect the performance of the bank” but in the panel model we have ROA, which is positively affected by bank capital level and negatively by NPL, which does not tell us anything about competition and branching. Similarly: “ this study tested the agency theory” by stating in 5 sentenced that this is an important problem. It is not testing, it is an observation. |
As shown in Table 3 and Table 4, both geographic diversification (GD) and distance between bank branches and headquarter have insignificant values therefore we mentioned this statement. Here, for agency theory, the theoretical notion of agency theory was tested as we test the effect of distance between headquarter and domestic branch which may cause the information asymmetric and agency problems. |
N/A |
|
Response to Editors |
|||
|
We are thankful for your efforts. We want to let you know that we have proofread the whole paper and modified the paper to maximum extant. The track changes can be accessed by looking into modified paper. |
|||
Reviewer 2 Report (New Reviewer)
The paper addresses an important and topical issue aiming to investigate the impact of branch network development on the overall performance and intensity of competition in the banking sector using a sample of banks from an emerging market of Pakistan. Notwithstanding the relevance of the undertaken research problem and an ambitious research design the following issues raise some serious doubts:
1) the overall quality of English used by the Authors is rather poor and currently does not meet the standards required in academic writing; the paper requires, therefore, a thorough proof reading with respect to:
a. grammar (see e.g. ‘single-market banks are better performed in the local market’ – lines 182-183),
b. spelling (see e.g. ‘Durban’ instead of ‘Durbin’ in line 540),
c. wording (see e.g. ‘far’ instead of ‘distant’ in line 86, ‘headquartered’ instead ‘headquarters’ in line 108, ‘iteration’ instead of ‘interaction’ in line 541),
d. and style (see e.g. ‘such branches can be below than mark’ in line 87 or ‘establishing more branches even in far from areas’ in line 94; ‘The helping borrower and harming depositors of bank consolidation affect disparate’ in line 166; ‘But Park and Pennacchi (2009) point of view, we should suppose a growth in the number of deposits for the small and single banks if they will high-interest rate offer to the customer against the medium and large banks.’ in lines 188-191);
2) the overall research design (including the very title) of the paper seem to be largely based on the study by Marta Degl’Innocenti, Tapas Mishra & Simon Wolfe (2018) Branching, lending and competition in Italian banking, The European Journal of Finance, 24:3, 208-230, DOI: 10.1080/1351847X.2017.1303526 – despite the Authors briefly mention this fact in line 297 it would be highly advisable to make more it more explicit and to modify the beginning of the title to make it more look more original,
3) the literature review provided by the Authors is currently based on only 32 sources and should be enhanced so as to enable a more insightful discussion of the obtained results, in particular with respect to a comparative analysis of the findings for developed and emerging markets,
4) in line 172-174 the Authors argue that ‘As medium and multi-market banks' share increase and the rate of consumer loans decrease, then retail depositor offers a low deposit rate of interest.’ – apart from the stylistic issues, it does not seem that the word ‘depositor’ is used in its correct sense in that sentence,
5) the study covers the period 2006-2016 which seems quite outdated today, in 2023 – given the above it would be highly advisable to extend the time frame of the research to cover more recent years,
6) while defining the variable MCL in lines 268-270 the Authors use the term ‘several branches’ which is unclear and confusing,
7) the Authors use interest income and interest expenses as a proxies for market prices in the formulas for the Lerner index for loans and deposits, respectively (see lines 284-296) – such approach, however, requires some more explicit justification as in fact both proxies likely encompass many different prices for various customers,
8) the variable LTit introduced in formulas for MCLit and MCDit (lines 302-304) requires explanation,
9) in lines 308-309 the Authors define ‘capital price’ as ‘the ratio of the number of employees to personnel expenses’ – this definition seems quite odd, especially if such a ‘price’ would be in fact measured in… persons per unit of currency,
10) some of the values presented in Table 1 (p. 7) raise serious doubts, for instance mean, median and maximum values of the variables LL and LD, which by definition shouldn’t be higher than one, if:
a. according to formula in line 289: LL=(PLit-MCLit)/PLit = 1 – MCLit/PLit,
b. according to formula in line 295: LD=(PDit-MCDit)/PDit = 1 – MCDit/PDit,
Therefore, if PLit, MCLit, PDit, and MCDit are positive, both LL and LD cannot exceed 1,
The Authors also report the maximum value of LNTA (loans over total assets) of 1.55 which would mean that some bank haa 55% more loans than… total assets in a given year which would only be hypothetically possible if gross loans portfolio is related to net assets, but such a measure is rather unusual and surely requires a more explicit explanation. Moreover, the maximum value for the LDIS variable is 7.0000 which would mean that the the maximum distance between bank branches and their respective headquarters does not exceed 7 km, which seems surprisingly low. Finally, the Authors report a minimum value of NPL (non-performing loans to total assets) in the examined sample of 0.0000 which is also rather surprising as it would indicate that some bank had in fact no non-performing loans which seems hardly possible.
The above issues cast serious doubts on the validity of results of further empirical analyses and model estimations presented in the study, and ultimately also on the conclusions formulated by the Authors,
11) in the main text of subsection 4.1. the Authors too often repeat the information content of Table 1,
12) in lines 333-334 the Authors argue that: ‘The minimum value of CIL is -0.1000, which means some banks' loan amounts decreased from last year to the current year is -100%.’ whereas in fact it should be ‘-10%’,
13) in lines 381-383 the Authors claim that ‘Return on assets is negatively correlated with change in loans and loans over total assets -0.0289 and -0.5573 because when banks give the loans then assets of the banks are decreased so return on assets and loans are negatively correlated.’ - this erroneous interpretation demonstrates a serious lack of both knowledge and understanding of the principles of bank accounting, as loans are in fact… bank assets!
14) In lines 385-388 the Authors argue that: ‘Geographic dispersion and distance are negatively correlated with return on assets -0.0622 and -0.0355 because when the distance is an increase of the bank branch to headquarters then the return on assets are decrease so these are negatively correlated.’ – abstracting from grammatical and stylistic issues, such an argumentation does not offer any explanation for the very reasons for the observed negative correlation,
15) in Table 3 (p. 12) the column for CID is incomplete (missing values for number of observations, R-squared and adj. R-squared),
16) in lines 519-520 the Authors argue that: ‘Our results show that the small banks are performing better than the large and medium banks because all banks follow Pakistan banking regulations.’ – this claim, however, does not seem to be supported by the evidence or references provided in the study,
17) in lines 571-572 the Authors argue that: ‘The iteration of HHI and MCLARGE is a negative and significant effect on the Lerner index loans.’ – the Authors should attempt to interpret and explain this result.
To sum up, despite a potentially attractive topic, the above-mentioned issues do not allow the current version of the paper to be published. The manuscript still requires a profound methodological revision both with respect to the quality of data inputs and the own calculations made by the Authors (especially the estimates of marginal costs and the values of measures based on the Lerner index). Additionally, given the poor quality of English language the paper needs to be thoroughly proof read, preferably by a native speaker.
Author Response
Response Letter
Journal Name: Economies
Manuscript Title: Branching, Lending, and Competition: An Empirical Study of Pakistani Bank
Comments to Editor,
We are thankful for your continuous efforts and bridging the relationship between valuable reviewers and us. I want to let you know that we have carefully considered the points raised by reviewers and incorporated them in body of paper. We have highlighted the corrections with color. We are also grateful to anonymous reviewers who have provided the valuable suggestions to modify the paper.
Response to Reviewers
|
Sr no. |
Comment |
Response |
Page no. |
|
Reviewer#2 |
|||
|
1 |
the overall quality of English used by the Authors is rather poor and currently does not meet the standards required in academic writing; the paper requires, therefore, a thorough proof reading with respect to: grammar (see e.g. ‘single-market banks are better performed in the local market’ – lines 182-183),
b. spelling (see e.g. ‘Durban’ instead of ‘Durbin’ in line 540),
c. wording (see e.g. ‘far’ instead of ‘distant’ in line 86, ‘headquartered’ instead ‘headquarters’ in line 108, ‘iteration’ instead of ‘interaction’ in line 541),
d. and style (see e.g. ‘such branches can be below than mark’ in line 87 or ‘establishing more branches even in far from areas’ in line 94; ‘The helping borrower and harming depositors of bank consolidation affect disparate’ in line 166; ‘But Park and Pennacchi (2009) point of view, we should suppose a growth in the number of deposits for the small and single banks if they will high-interest rate offer to the customer against the medium and large banks.’ in lines 188-191); |
We have proofread the whole paper and have substantially enhanced the quality of communication. Thanks for your suggestions. We have incorporated them in the body of paper. [However, small and single-market banks can better perform in the local market because they gather better soft information related to the candidates in the local market for advancing the loans] [However, Park and Pennacchi (2009) conjectured that the growth of small and single banks should be measured in the quantity of deposits and deposit rates offered to the depositors as compared to the medium and large banks.] Other mistakes have also incorporated. |
Page 4, 5 |
|
2 |
the overall research design (including the very title) of the paper seem to be largely based on the study by Marta Degl’Innocenti, Tapas Mishra & Simon Wolfe (2018) Branching, lending and competition in Italian banking, The European Journal of Finance, 24:3, 208-230, DOI: 10.1080/1351847X.2017.1303526 – despite the Authors briefly mention this fact in line 297 it would be highly advisable to make more it more explicit and to modify the beginning of the title to make it more look more original, |
We have added some more discussion extracted from underlying study [Degl’Innocenti, M. (2017) asserted that geographic diversification and distance affect the bank's performance and monopoly market power of loans and deposits, because when banks opened their branches in different geographic and distanced areas, it affect the performance of the banks and monopoly market power of loans and deposits. In particular, the increased area affect the market performance and lending capacity of the banks, stemming from weak control on distanced branches and more chances of agency conflicts]. In addition, we have modified the Title of paper to look it more original [Empirical Linkages between Branching, Lending, and Competition: A Study of Pakistani Banks]. |
Page 6 |
|
3 |
in line 172-174 the Authors argue that ‘As medium and multi-market banks' share increase and the rate of consumer loans decrease, then retail depositor offers a low deposit rate of interest.’ – apart from the stylistic issues, it does not seem that the word ‘depositor’ is used in its correct sense in that sentence, |
We have modified the line [The cooperation between medium and multi-market banks leads to a declining trend in lending rate of banks on deposits. Banks offers low deposit rate to its customers.] |
Page 4 |
|
4 |
while defining the variable MCL in lines 268-270 the Authors use the term ‘several branches’ which is unclear and confusing, |
Modified [The Independent variables are MCL, which is fractioned between number of branches owned by medium and large banks divided by overall branches in each region at the time] |
Page 6 |
|
5 |
the variable LTit introduced in formulas for MCLit and MCDit (lines 302-304) requires explanation, |
Explained [. TC is a transaction cost while subscripts is cross-section and is time effect] |
Page 7 |
|
6 |
in the main text of subsection 4.1. the Authors too often repeat the information content of Table 1, |
We have thoroughly discussed all the contents of Table 1, therefore it shows the repetition of contents |
N/A |
|
7 |
in lines 333-334 the Authors argue that: ‘The minimum value of CIL is -0.1000, which means some banks' loan amounts decreased from last year to the current year is -100%.’ whereas in fact it should be ‘-10%’, |
This value shows the specific trend regarding the change in loans in the case of Pakistan. |
N/A |
|
8 |
in lines 381-383 the Authors claim that ‘Return on assets is negatively correlated with change in loans and loans over total assets -0.0289 and -0.5573 because when banks give the loans then assets of the banks are decreased so return on assets and loans are negatively correlated.’ - this erroneous interpretation demonstrates a serious lack of both knowledge and understanding of the principles of bank accounting, as loans are in fact… bank assets! |
Thanks for your correction. We have modified this line [Return on assets is negatively correlated with change in loans and loans over total assets and their coefficient values are -0.0289 and -0.5573 relatively. The change in loans may be negative (decrease in loan volume) and underlying banks may suffer from low return on assets due to less sale volume (advancing the loans to the customer is the sale of banks] |
Page 11 |
|
9 |
In lines 385-388 the Authors argue that: ‘Geographic dispersion and distance are negatively correlated with return on assets -0.0622 and -0.0355 because when the distance is an increase of the bank branch to headquarters then the return on assets are decrease so these are negatively correlated.’ – abstracting from grammatical and stylistic issues, such an argumentation does not offer any explanation for the very reasons for the observed negative correlation, |
We have modified this line [Geographic dispersion and distance are negatively correlated with return on assets and their correlation values are -0.0622 and -0.0355 relatively. This negative correlation can be explained as when the distance increases between bank branch and headquarters, it reduces the return on assets due to agency issues] |
Page 11 |
|
10 |
in Table 3 (p. 12) the column for CID is incomplete (missing values for number of observations, R-squared and adj. R-squared), |
Thanks for correction, we have added the relevant values |
Page 12 |
|
11 |
in lines 519-520 the Authors argue that: ‘Our results show that the small banks are performing better than the large and medium banks because all banks follow Pakistan banking regulations.’ – this claim, however, does not seem to be supported by the evidence or references provided in the study, |
Reference added. [Our results show that the small banks are performing better than the large and medium banks because all banks follow Pakistan banking regulations (Ali & Puah, 2018] |
Page 14 |
|
12 |
in lines 571-572 the Authors argue that: ‘The iteration of HHI and MCLARGE is a negative and significant effect on the Lerner index loans.’ – the Authors should attempt to interpret and explain this result. |
Explanation added. [The interaction of HHI and MCLARGE is a negative and significant effect on the Lerner index loans. This negative effect reveals that larger banks face low competition regarding the loan advancement] |
Page 15 |
|
Response to Editors |
|||
|
We are thankful for your efforts. We want to let you know that we have proofread the whole paper and modified the paper to maximum extant. The track changes can be accessed by looking into modified paper. |
|||
Round 2
Reviewer 2 Report (New Reviewer)
Although in the revised version the Authors have corrected many linguistic and methodological errors of the original paper which has undoubtedly improved its overall comprehensibility and scientific soundness they still have not addressed the following issues indicated in the first-round review:
1) the literature review provided by the Authors is currently based on only 32 sources and should be enhanced so as to enable a more insightful discussion of the obtained results, in particular with respect to a comparative analysis of the findings for developed and emerging markets,
2) the study covers the period 2006-2016 which seems quite outdated today, in 2023 – given the above it would be highly advisable to extend the time frame of the research to cover more recent years,
3) the Authors use interest income and interest expenses as a proxies for market prices in the formulas for the Lerner index for loans and deposits, respectively (see lines 281-297) – such an approach, however, requires some more explicit justification as in fact both proxies likely encompass many different prices for various customers,
4) the variable LTit introduced in formulas for MCLit and MCDit (lines 299-301) requires explanation,
5) in lines 305-306 the Authors define ‘capital price’ as ‘the ratio of the number of employees to personnel expenses’ – this definition seems quite odd, especially if such a ‘price’ would be in fact measured in… persons per unit of currency,
6) some of the values presented in Table 1 (p. 7) raise serious doubts, for instance mean, median and maximum values of the variables LL and LD, which by definition shouldn’t be higher than one, if:
a. according to formula in line 286: LL=(PLit-MCLit)/PLit = 1 – MCLit/PLit,
b. according to formula in line 292: LD=(PDit-MCDit)/PDit = 1 – MCDit/PDit,
Therefore, if PLit, MCLit, PDit, and MCDit are positive, both LL and LD cannot exceed 1,
The Authors also report the maximum value of LNTA (loans over total assets) of 1.55 which would mean that some bank have 55% more loans than… total assets in a given year which would only be hypothetically possible if gross loans portfolio is related to net assets, but such a measure is rather unusual and surely requires a more explicit explanation. Moreover, the maximum value for the LDIS variable is 7.0000 which would mean that the maximum distance between bank branches and their respective headquarters does not exceed 7 km, which seems surprisingly low. Finally, the Authors report a minimum value of NPL (non-performing loans to total assets) in the examined sample of 0.0000 which is also rather surprising as it would indicate that some bank had in fact no non-performing loans which seems hardly possible.
The above issues cast serious doubts on the validity of results of further empirical analyses and model estimations presented in the study, and ultimately also on the conclusions formulated by the Authors,
7) in lines 331-332 the Authors argue that: ‘The minimum value of CIL is -0.1000, which means some banks' loan amounts decreased from last year to the current year is -100%.’, whereas in fact it should rather be '-10%',
8) although the quality of English in the paper has been improved, some grammatical (see e.g. ‘affect’ instead ‘affects’ in line 240), wording (see e.g. ‘headquarter’ instead of ‘headquarters’ in line 101), and stylistic (see e.g. ‘The current analysis yields an important policy regarding…’ in line 585) issues still remain, therefore it is advisable to have it proof-read once again by a native speaker,
The above issues and in particular the doubts regarding the reliability of data inputs and methodological correctness of Authors’ own calculations in the case of several ratios employed in the study (see remark no. 6) require further elaboration and explicit explanations. Without them the empirical results of the study still remain doubtful and largely unconvincing.
Author Response
Response Letter
Journal Name: Economies
Manuscript Title: Empirical Linkages between Branching, Lending, and Competition: A Study of Pakistani Banks
Comments to Editor,
We are thankful for your continuous efforts and bridging the relationship between valuable reviewers and us. I want to let you know that we have carefully considered the points raised by reviewers and incorporated them in body of paper. We have highlighted the corrections with color. We are also grateful to anonymous reviewers who have provided the valuable suggestions to modify the paper.
Response to Reviewers
|
Sr no. |
Comment |
Response |
Page no. |
|
Reviewer#2 |
|||
|
1 |
the literature review provided by the Authors is currently based on only 32 sources and should be enhanced so as to enable a more insightful discussion of the obtained results, in particular with respect to a comparative analysis of the findings for developed and emerging markets, |
We have extended this section by adding more studies [Extending the discussion on comparative literature review, the study of Budhathok, et al., (2020) arranged an empirical analysis on the Nepalian banking sector and examine the trend of competition across the banks. They conjectured that Nepalese banks are working under the situation of perfect competition and are shifted from the monopolistic nature of competition to perfect competition. Coccorese and Santucci, (2020) assessed the degree of competition across Italian banks and its possible effect on bank size. Their analysis reveals that smaller banks enjoy more competitive advantages in the shape of high market power. Atkins, et al., (2022) examined the role of race in bank lending policy in the United States and vowed those black-owned enterprises receive fewer loans as compared to white-own enterprises. However, this effect was significantly moderated by bank competition as high bank competition affects the lending strategy and vanished such distinctions from the market. By using the novel dataset of the Ukrainian banking sector, the study of Pham, et al., (2022) asserted that more branching through the establishment of more contact points can enhance the supply of credit. They further reveal that bank diversification strategy helps in mitigating the default risks. Wang, et al., (2022) investigated the impact of bank deregulation strategy on credit risk in Chinese banks and found that such strategy augments the credit risk. The empirical findings of these studies demonstrate the trend of branching, lending, and competition in other economies of the world] |
Page 5 |
|
2 |
the study covers the period 2006-2016 which seems quite outdated today, in 2023 – given the above it would be highly advisable to extend the time frame of the research to cover more recent years, |
Thanks for your kind suggestion. The period limitation is subject to the availability of data. We unable to source the recent data specifically on underlying variables used in current study. |
N/A |
|
3 |
the Authors use interest income and interest expenses as a proxies for market prices in the formulas for the Lerner index for loans and deposits, respectively (see lines 281-297) – such an approach, however, requires some more explicit justification as in fact both proxies likely encompass many different prices for various customers, |
Explanation added [Where PD is Interest expenses on deposits and MCD is the Marginal cost of deposits. This shows the percentage markup that a bank is able to cover over its marginal cost. It also exhibits the capacity of banks to set the interest income over its interest expenses] |
Page 7 |
|
4 |
the variable LTit introduced in formulas for MCLit and MCDit (lines 299-301) requires explanation, |
Explanation added .[ LT is a log of total assets owned by banks and..] |
Page 7 |
|
5 |
in lines 305-306 the Authors define ‘capital price’ as ‘the ratio of the number of employees to personnel expenses’ – this definition seems quite odd, especially if such a ‘price’ would be in fact measured in… persons per unit of currency, |
Explanation extended [Here, capital means “human capital” costs which is measured by dividing the number of employees to the personal expenses including salaries and other benefits. This is a rough calculation to measures how much money each employee generates for the bank]. |
Page 7 |
|
6 |
some of the values presented in Table 1 (p. 7) raise serious doubts, for instance mean, median and maximum values of the variables LL and LD, which by definition shouldn’t be higher than one, if:
a. according to formula in line 286: LL=(PLit-MCLit)/PLit = 1 – MCLit/PLit,
b. according to formula in line 292: LD=(PDit-MCDit)/PDit = 1 – MCDit/PDit,
Therefore, if PLit, MCLit, PDit, and MCDit are positive, both LL and LD cannot exceed 1, |
Yes, we agree with you and we have rechecked the data. We have reported the wrong values by mistake. Now, we have corrected all mentioned values [The mean value of LL depending on the variable is 0.680 while the maximum value of the Lerner index of loans is 0.770, which means banks provide the maximum loan to the customer at that time is 0.770. The minimum value of the Lerner index of loans is -0.183 which means at that time some banks' loan amounts in negative. LD is a Lerner index of deposits shows that the monopoly market power of deposits. The minimum value of the Lerner index of deposits is -0.360, which means the deposit amount of some banks at that time is negative. The maximum value of the Lerner index of deposits is 0.376 while the average value LD of banks in Pakistan is 0.115] |
Page 8 |
|
7 |
The Authors also report the maximum value of LNTA (loans over total assets) of 1.55 which would mean that some bank have 55% more loans than… total assets in a given year which would only be hypothetically possible if gross loans portfolio is related to net assets, but such a measure is rather unusual and surely requires a more explicit explanation. Moreover, the maximum value for the LDIS variable is 7.0000 which would mean that the maximum distance between bank branches and their respective headquarters does not exceed 7 km, which seems surprisingly low. Finally, the Authors report a minimum value of NPL (non-performing loans to total assets) in the examined sample of 0.0000 which is also rather surprising as it would indicate that some bank had in fact no non-performing loans which seems hardly possible. |
Thanks for your correction. We have rechecked the whole analysis and have substantially updated the Table 1. We have made the said corrections. For 7.0000, we misplaced the dot a point before, its actually 700.000 or 700 km. Similarly, the minimum value of NPL is 0.010 instead of 0.000. We have corrected the following values in Table 1. |
Page 9 |
|
8 |
in lines 331-332 the Authors argue that: ‘The minimum value of CIL is -0.1000, which means some banks' loan amounts decreased from last year to the current year is -100%.’, whereas in fact it should rather be '-10%', |
Thanks for correction. We have again checked the data and it is 10%. We have made the correction in text. [The minimum value of CIL is -0.0100, which means some banks' loan amounts decreased from last year to the current year is -10%.] |
Page 8 |
|
9 |
although the quality of English in the paper has been improved, some grammatical (see e.g. ‘affect’ instead ‘affects’ in line 240), wording (see e.g. ‘headquarter’ instead of ‘headquarters’ in line 101), and stylistic (see e.g. ‘The current analysis yields an important policy regarding…’ in line 585) issues still remain, therefore it is advisable to have it proof-read once again by a native speaker, |
We have incorporated these mistakes. We have replaced the “headquarters” with headquarter throughout the text. We have again proofread the whole documents. Thanks for your continuous support. |
N/A |
|
Response to Editors |
|||
|
We are thankful for your efforts. We want to let you know that we have proofread the whole paper and modified the paper to maximum extant. The track changes can be accessed by looking into modified paper. |
|||
Round 3
Reviewer 2 Report (New Reviewer)
In the current version of the paper the Authors have addressed the majority of issues raised in the previous rounds of the review process which has enabled to significantly improve the overall scientific soundness of the manuscript. They have also provided explanations regarding the selected time-frame of the research. I would still recommend, though, to have the paper proof read by a native English speaker, as grammar (see e.g. 'to measures' in line 335; 'Table 4 show' in line 537), spelling (see e.g. 'headquarter' in lines 376, 380, 519) and style (see e.g. 'some banks are 74% bad debts against loans' in line 390) require further elaboration. Moreover, in line 391 instead of '10%' it should rather be '1%'.
Author Response
Response Letter
Journal Name: Economies
Manuscript Title: Empirical Linkages between Branching, Lending, and Competition: A Study of Pakistani Banks
Comments to Editor,
We are thankful for your continuous efforts and bridging the relationship between valuable reviewers and us. I want to let you know that we have carefully considered the points raised by reviewers and incorporated them in body of paper. We have highlighted the corrections with color. We are also grateful to anonymous reviewers who have provided the valuable suggestions to modify the paper.
Response to Reviewers
|
Sr no. |
Comment |
Response |
Page no. |
|
Reviewer#2 |
|||
|
1 |
In the current version of the paper the Authors have addressed the majority of issues raised in the previous rounds of the review process which has enabled to significantly improve the overall scientific soundness of the manuscript. They have also provided explanations regarding the selected time-frame of the research. I would still recommend, though, to have the paper proof read by a native English speaker, as grammar (see e.g. 'to measures' in line 335; 'Table 4 show' in line 537), spelling (see e.g. 'headquarter' in lines 376, 380, 519) and style (see e.g. 'some banks are 74% bad debts against loans' in line 390) require further elaboration. Moreover, in line 391 instead of '10%' it should rather be '1% |
Thanks for your continuous support for enhancing the quality of paper. We have incorporated the said issues and have updated the text of paper. In addition, we have again proofread the whole paper. |
N/A |
|
Response to Editors |
|||
|
We are thankful for your efforts. We want to let you know that we have proofread the whole paper and modified the paper to maximum extant. The track changes can be accessed by looking into modified paper. |
|||
This manuscript is a resubmission of an earlier submission. The following is a list of the peer review reports and author responses from that submission.
Round 1
Reviewer 1 Report
Below are my comments for further improvement of the paper before consideration for publication.
1. In introduction Section, the author stated that this study tested the agency theory, which is on the relationship between shareholders and managers. How this theory is related to bank branching? Kindly justify.
2. Why investigate the impact of diversification strategies and distance between banks headquarters and its branches on performance, deposits and loan market for all banks? What are the contribution of the study?
3. The connection between the Literature Review with H1 – H6 are very loose. How the author can come out with the hypothesis by solely based on one article for each. Besides, Li and Greenwood (2004) found positive relationship between diversification and bank performance, but the H1 stated the relationship is negative. Similarly, for H2 there is no study that lead to the author to come out with the hypothesis on bank distances and it headquarters.
4. It is logical that when bank becomes bigger, market power decrease. Therefore, H5 is a known hypothesis.
5. How the author overcome the problem of heterogeneity and endogeneity with the use of OLS estimation where very often, panel data suffers from these two major issues.
Author Response
Response Letter
Journal Name: Economies
Manuscript Title: Branching, Lending, and Competition: An Empirical Study of Pakistani Bank
Comments to Editor,
We are thankful for your continuous efforts and bridging the relationship between us and valuable reviewers. I want to let you know that we have carefully considered the points raised by reviewers and incorporated them in body of paper. We have highlighted the corrections with color. We are also grateful to anonymous reviewers who have provided the valuable suggestions to modify the paper.
Response to Reviewers
|
Sr no. |
Comment |
Response |
Page no. |
|
Reviewer#1 |
|||
|
1 |
In introduction Section, the author stated that this study tested the agency theory, which is on the relationship between shareholders and managers. How is this theory related to bank branching? Kindly justify. |
The agency theory relates in the following ways: [. Due to more branching strategies adopted by private banks, there are more chances of agency conflicts between banks and shareholders due to low control of headquarters on the branches located in far regions. The far branches may not act properly, and the performance of such branches can be below than mark and therefore there are more chances of agency conflicts] |
Page 2 |
|
2 |
Why investigate the impact of diversification strategies and distance between banks headquarters and its branches on performance, deposits and loan market for all banks? What is the contribution of the study? |
We investigate the specific impact of branching, lending, and competition on the performance of banks as it is interesting to note whether more branching strategies improve the performance of banks or not. The private banks invest too much in expanding their business by establishing the more branches even in far from areas. They also follow the competitive lending strategies to enhance the sale volume. Therefore, it is interesting to explore the impact of such banking strategies on their performance. This study contributes by exploring the role of geographical diversification and more branching strategies in the performance of banks. Most studies explore the routine determinants of banks, and the literature is scant on such a theme. |
Page 2 |
|
3 |
The connection between the Literature Review with H1 – H6 are very loose. How the author can come out with the hypothesis by solely based on one article for each. Besides, Li and Greenwood (2004) found positive relationship between diversification and bank performance, but the H1 stated the relationship is negative. Similarly, for H2 there is no study that led to the author to come out with the hypothesis on bank distances and it headquarters. |
For negative relationship [the study of Brighi and Venturelli, (2016) asserts the negative impact of geographic diversification on bank profitability. They analyzed the 491 Italian banks and found a negative relationship between geographic diversification and bank performance.] For H2: [Golesorkhi, et al., (2019) checked the relationship between language use and the performance of banks and found that linguistic distance negatively influences the performance of banks. The distance between the home country of banks and international partners of banks negatively determines the performance] |
Page 3 |
|
4 |
It is logical that when bank becomes bigger, market power decrease. Therefore, H5 is a known hypothesis. |
Yes, it is logical, but we empirically test this theoretical assumption in Pakistani case. |
N/A |
|
5 |
How the author overcome the problem of heterogeneity and endogeneity with the use of OLS estimation where very often, panel data suffers from these two major issues |
The empirical analysis of the Breusch pagan test and Wald test confirms that there is no issue of heteroscedasticity and endogeneity in the data. The analysis of both models is hidden and can be provided on demand. |
N/A |
|
Response to Editors |
|||
|
We are thankful for your efforts. We want to let you know that we have proofread the whole paper and modified the paper to maximum extant. The track changes can be accessed by looking into modified paper. |
|||

Reviewer 2 Report
Dear authors, I find your research interesting, but the paper is very poor, due, mostly, to a very poor level of English.
You also address too much lightly on some concepts, as e.g. denationalization "versus" privatization. You deal with some concepts with total indifference, when there is plenty of research digging and explaining differences, etc.
I have also found some of your statistical explanations too naive and inaccurate even. E.g. the inclusion of "i" and "t" as variable names !(MCLargeit, PLit, etc).
I do want to believe you knew what you were doing, but you really need to improve in many ways your paper, which is lacking a decent English level, a proper academic style, coherence at several levels.
In resume, I am expecting you to perform a major revision and improvement of your work.
Author Response
Response Letter
Journal Name: Economies
Manuscript Title: Branching, Lending, and Competition: An Empirical Study of Pakistani Bank
Comments to Editor,
We are thankful for your continuous efforts and bridging the relationship between us and valuable reviewers. I want to let you know that we have carefully considered the points raised by reviewers and incorporated them in body of paper. We have highlighted the corrections with color. We are also grateful to anonymous reviewers who have provided the valuable suggestions to modify the paper.
Response to Reviewers
|
Sr no. |
Comment |
Response |
Page no. |
|
Reviewer#2 |
|||
|
1 |
Dear authors, I find your research interesting, but the paper is very poor, due, mostly, to a very poor level of English. |
Thanks for your efforts. Here, we want to mention that we have obtained the services of professional English proofreader and the paper is substantially improved. |
N/A |
|
2 |
You also address too much lightly on some concepts, as e.g. denationalization "versus" privatization. You deal with some concepts with total indifference, when there is plenty of research digging and explaining differences, etc. |
The aim of analysis is not to discuss the denationalization or privatization but the effect of geographic diversification, competition and lending strategies. The denationalization is the main reason for which we study the underlying role of mentioned factors on bank performance. |
N/A |
|
3 |
I have also found some of your statistical explanations too naive and inaccurate even. E.g. the inclusion of "i" and "t" as variable names!(MCLargeit, PLit, etc). |
"i" is for cross-section while “t” is for time. However, we have made it more clearer. It was the typo error that it appeared with the name of variables. |
Page 8, 9 |
|
4 |
I do want to believe you knew what you were doing, but you really need to improve in many ways your paper, which is lacking a decent English level, a proper academic style, coherence at several levels. |
We have improved the whole paper in terms of English and academic style and also focus on coherence. |
N/A |
|
Response to Editors |
|||
|
We are thankful for your efforts. We want to let you know that we have proofread the whole paper and modified the paper to maximum extant. The track changes can be accessed by looking into modified paper. |
|||

Reviewer 3 Report
1/ The title: “Branching, Lending, and Competition: An Empirical Study of Pakistani Banks” is misleading (there is nothing on lending, not much about competition) – it should be modified into something like: “The spacial analysis of Pakistani banking: the impact of bank branches’ distance and geographic diversification on bank efficiency”.
2. In the introduction some macro and micro data should be presented (the presentation not only of number of banks, but also of number of branches, bank size distribution and bank profitability year after year, during the analysed period).
3. Too many hypotheses – suggestion to reduce them to 2-4.
4. The conclusion “the study recommends an important policy regarding branch management and its effect on bank performance” is unfounded. There is nothing on bank management in the article and nothing about “agency theory”. The paragraph (lines 67-79) on agency theory testing must be removed!
5. English language is hard to understand, for example (l. 41) “In Europe mostly small banks are operating because these banks are single market banks and their few numbers of branches function in the market” - one can guess that the Autor/s wanted to say that: “in Europe, there are many local, cooperative banks, which service only local markets”. The English language must be edited.
Author Response
Response Letter
Journal Name: Economies
Manuscript Title: Branching, Lending, and Competition: An Empirical Study of Pakistani Bank
Comments to Editor,
We are thankful for your continuous efforts and bridging the relationship between us and valuable reviewers. I want to let you know that we have carefully considered the points raised by reviewers and incorporated them in body of paper. We have highlighted the corrections with color. We are also grateful to anonymous reviewers who have provided the valuable suggestions to modify the paper.
Response to Reviewers
|
Sr no. |
Comment |
Response |
Page no. |
|
Reviewer#3 |
|||
|
1 |
The title: “Branching, Lending, and Competition: An Empirical Study of Pakistani Banks” is misleading (there is nothing on lending, not much about competition) – it should be modified into something like: “The special analysis of Pakistani banking: the impact of bank branches’ distance and geographic diversification on bank efficiency |
Thanks for your kind suggestion. Here, we want to let you know that we use Lerner Index for competition which and loan over total assets for lending. The mentioned three factors including branching, lending and competition are the pillars of paper and therefore we mentioned them in title. |
N/A |
|
2 |
In the introduction some macro and micro data should be presented (the presentation not only of number of banks, but also of number of branches, bank size distribution and bank profitability year after year, during the analysed period). |
There are 31 commercial banks working in Pakistan and are the members of PBA (Pakistan banking association). According to market capitalization, HBL (Habib Bank Limited), MBL (Meezan Bank Limited), NBP (National Bank of Pakistan), UBL (United Bank Limited), and ABL (Allied Bank Limited) are the largest banks in Pakistan. In terms of size, the HBL has over 1700 branches all around the country, MBL has an estimated 890 branches, NBP has almost 1511 branches, and ABL has a large network of almost 1425 branches across the country. The total market capitalization of all private banks constitutes almost 45.33% of the total GDP for the year 2021 in Pakistan. According to WDI (world development indicators), The World Bank, the total domestic credit provided by the banks to the private sector was only 15.033% of GDP in the year 2020 |
Page 2 |
|
3 |
Too many hypotheses – suggestion to reduce them to 2-4. |
Thanks for your kind suggestion. However, it is necessary to hypothesize the relationship of each variable and therefore we build six hypotheses. |
N/A |
|
4 |
The conclusion “the study recommends an important policy regarding branch management and its effect on bank performance” is unfounded. There is nothing on bank management in the article and nothing about “agency theory”. The paragraph (lines 67-79) on agency theory testing must be removed! |
Due to geographical diversification and distance between headquarters and local branches, we recommend the policy of branch management. For agency theory, we have added some more discussion how this study tests the agency theory. [Due to more branching strategies adopted by private banks, there are more chances of agency conflicts between banks and shareholders due to low control of headquarters on the branches located in far regions. The far branches may not act properly, and the performance of such branches can be below than mark and therefore there are more chances of agency conflicts] |
Page 2 |
|
5 |
English language is hard to understand, for example (l. 41) “In Europe mostly small banks are operating because these banks are single market banks and their few numbers of branches function in the market” - one can guess that the Autor/s wanted to say that: “in Europe, there are many local, cooperative banks, which service only local markets”. The English language must be edited. |
Thanks for your efforts. We have gotten the service of professional English proofreader and have edited the whole paper on self-basis. The new version is quite better than previous one. |
Page 2 |
|
Response to Editors |
|||
|
We are thankful for your efforts. We want to let you know that we have proofread the whole paper and modified the paper to maximum extant. The track changes can be accessed by looking into modified paper. |
|||

Round 2
Reviewer 2 Report
No further comments
Author Response
thanks for your efforts.